# The Nuclear Speckles Protein SRRM2 Is Exposed on the Surface of Cancer Cells

**DOI:** 10.3390/cells13181563

**Published:** 2024-09-17

**Authors:** Markus Kellner, Julia Hörmann, Susanne Fackler, Yuanyu Hu, Tielin Zhou, Lin Lu, Ibrahim Ilik, Tugce Aktas, Regina Feederle, Stefanie M. Hauck, Olivier Gires, Kathrin Gärtner, Lietao Li, Reinhard Zeidler

**Affiliations:** 1Institute of Structural Biology, Helmholtz Zentrum München, German Research Center for Environmental Health, Feodor-Lynen-Str. 21, 81377 Munich, Germany; 2Eximmium Biotechnologies GmbH, 81377 Munich, Germany; 3Zeno Therapeutics Pte. Ltd., 600 North Bridge Road, Singapore 188778, Singapore; 4Otto Warburg Laboratories, 14195 Berlin, Germany; 5Core Facility Monoclonal Antibodies, Helmholtz Zentrum München, German Research Center for Environmental Health, 85764 Neuherberg, Germany; 6Proteomics and Metabolomics Core, Helmholtz Zentrum München, German Research Center for Environmental Health, 85764 Neuherberg, Germany; 7Department of Otorhinolaryngology, LMU University Hospital, 81377 Munich, Germany

**Keywords:** SRRM2, target identification, cancer target, therapeutic antibodies

## Abstract

The membrane composition of extracellular vesicles (EVs) largely reflects that of the plasma membrane of the cell of origin. We therefore hypothesized that EVs could be used for immunizations to generate monoclonal antibodies against well-known tumor antigens but possibly also against hitherto unknown tumor-associated target molecules. From an immunization experiment, we obtained a monoclonal antibody specific for SRRM2, an RNA-binding protein involved in splicing and a major component of nuclear speckles. Here, we used this antibody to demonstrate that SRRM2 is exposed on the surface of most cancer cell lines from various entities and, even more important, on cancer cells in vivo. Moreover, we demonstrated that SRRM2-specific CAR-T cells are functional in vitro and in vivo. Collectively, we identified SRRM2 as a promising new target molecule exposed on the cancer cell surface and showed that our SRRM2-specific antibody can be used as a basis for the development of new targeted cancer therapies.

## 1. Introduction

Personalized targeted therapies hold great promise for the development of efficient and well-tolerated treatment regimens for life-threatening diseases like cancer. Therapeutic monoclonal antibodies are one such success story in modern medicine, that regularly show amazing clinical data, at least in a fraction of cancer patients. Faster developments and a broader application of targeted immunotherapies—above all for the treatment of solid tumors—is hampered by the small number of known suitable target molecules that are exclusively or predominantly expressed on the surface of cancer cells. Hence, there is an urgent need for the continuous identification of such new target molecules for the development of innovative therapies to substantially increase the number of cancer patients eligible to targeted therapies.

Extracellular vesicles (EVs) are a cluster of nanosized vesicles delimited by a lipid bilayer that are released by almost all types of cells, including cancer cells. EVs are rich in membrane proteins and, when released from cancer cells, carry established tumor-associated antigens (TAAs) like Her2, CEA, EpCAM, etc. [1,2,3]. Since, in terms of their composition, EVs largely represent the cells of origin, we speculated that cancer-derived EVs also carry proteins not yet recognized as TAAs, thus constituting a valuable tool for the identification of new druggable target molecules on the surface of cancer cells. To prove this hypothesis, we established, throughout the last years, a proprietary technology for the generation of monoclonal antibodies relying on the use of cancer-derived EVs for immunizations. Immunizations with EVs isolated from permanent human cancer cell lines of different origins yielded many antibodies targeting membrane proteins including established TAAs like EpCAM and Her2/neu, but also against proteins which have hitherto never been described to be exposed on the cell surface.

Unexpectedly, we obtained an antibody specific for SRRM2 (serine/arginine repetitive matrix protein 2; aka SRM300) a large, mostly unstructured serine/arginine-rich (SR) protein which is a component of spliceosomal complexes [4]. SRRM2 is one of the core scaffold proteins required for the proper formation of nuclear speckles [5], i.e., membrane-less nuclear organelles, where mRNA maturation and splicing take place but whose exact function remain elusive. SR proteins play a role in affecting alternative splice sites in vitro and in vivo [6,7], and SRRM2 is also known to play a central role in mRNA splicing [4,8]. As a member of the SR family, SRRM2 contains an N-terminal RNA recognition motif and a large serine/arginine-rich C-terminal low-complexity intrinsically disordered region (IDR) [5,9] thought to mediate protein–protein interaction and liquid–liquid phase separation (LLPS). Interestingly, SRRM2 has been found to accumulate in neuron cytoplasm in Alzheimer’s disease (AD), frontotemporal dementia (FTD), and other neurodegenerative diseases [10,11,12].

Aberrant translocation of normally intracellular proteins to the surface of cancer cells is a well-known yet poorly understood and emphasized phenomenon [13]. One prominent example is BiP/Grp78/HSPA5, a member of the heat shock protein (hsp)70 family of chaperones. While in normal cells BiP is endoplasmic reticulum-resident, it is also exposed on the surface of many types of cancers, where it probably exerts a disparate function that contributes to tumor progression [14,15]. Other examples for mislocated proteins in cancer cells constitute the phosphatase PRL3 [16], hsp90 ([17] for review), nucleolin ([18] for review), and alpha-enolase ENO1 [19], among others. Overall, there is an ever-increasing interest in the identification of such multifunctional proteins which translocate to the outer surface of cancer cells. As this mechanism seems very tumor-specific, the identified proteins are considered promising and attractive new target molecules for the development of novel specific cancer therapies.

Here, we describe the generation of an SRRM2-specific antibody using EVs derived from cancer cell lines for immunizations, demonstrating that SRRM2 is released from cancer cells via EVs. Additionally, we show that SRRM2 is exposed on the surface of human cancer cells from different entities and constitutes a novel appealing cancer-associated target molecule, opening the way towards the development of novel SRRM2-targeting cancer therapies.

## 2. Material and Methods

### 2.1. Cells and Antibodies

Cell lines were grown in DMEM supplemented with 7% FCS at 37 °C in a humified atmosphere with 5% CO_2_. SKOV-3 (ovarian cancer; ATCC HTB-77), UWB1.289 (ovarian cancer; ATCC CRL-2945), T-47D (breast cancer; ATCC HTB-133), HeLa (cervical cancer; ATCC CCL-2), SkBr3 (breast cancer; ATTCC HTB-30), and Capan-1 (pancreatic cancer; ATCC HTB-79) were obtained from ATCC; PCI-1 is a human hypopharyngeal cancer cell line and a kind gift from Prof. T. Whiteside (Pittsburgh, PA, USA). The generation of HAP1 cell lines carrying GFP-tagged SRRM2 mutants has been described recently [5]. Primary PBMCs from anonymized volunteer blood donors were purchased from the Division of Transfusion Medicine of the LMU University Hospital Munich with informed consent of the donors. Primary hepatocytes were obtained from HTCR Services GmbH (Munich, Germany). The following primary antibodies were used: EpCAM (clone C215 [20]; a kind gift of Dr. H. Lindhofer, Munich, Germany) and SRRM2 (Abcam (Cambridge, UK); clone 122719, Biozol (Eching, Germany); catalog no. 9206; ThermoFisher Scientific (Waltham, MA, USA); clone PA5-66827, and Sigma Aldrich (Deisenhof, Germany); clone SC-35). CD3, CD4, CD8, CD45, and CD63-HRP antibodies were purchased from ThermoFisher Scientific (Darmstadt, Germany). The anti-His antibody and isotype control antibody were obtained from the Core Facility Monoclonal Antibodies, Helmholtz Munich. Fluorochrome and HRP-labeled secondary antibodies were purchased from Jackson ImmunoResearch (Cambridgeshire, UK).

### 2.2. Generation of Antibodies

A Lou/c rat was immunized with a mixture of extracellular vesicles, which were derived from SKOV-3, Capan-1, T-47D, and HeLa cells and isolated from conditioned supernatants by serial centrifugation. Briefly, conditioned FCS-free supernatants were collected, subjected to repeated centrifugations at increasing centrifugal force (10 min at 300× *g*, 4 °C, and 20 min at 5000× *g*, 4 °C), filtrated (pore size 0.45 µm), and, finally, precipitated at 100,000× g, 4 °C for 2 h in a SW28 rotor. Pelleted vesicles were resuspended in 100 µL of PBS and injected i.p. and s.c. with CpG as adjuvant. A boost injection was given five months later, and spleen cells were fused with myeloma cell line P3×63Ag8.653 (ATCC, CRL-1580). Hybridoma supernatants were screened ten days later for IgG production, and positive clones were further expanded and subcloned at least twice by limiting dilution to obtain stable monoclonal cell lines. Experiments in this work were performed with clone OCAM 23A7/EX-02 (rat IgG2b).

### 2.3. Immunoblotting and Immunoprecipitation

Cells were lysed in ice-cold RIPA lysis buffer (0.1% SDS, 50 mM Tris-HCl pH 8.0, 0.5% DOC, 1% NP-40, 150 mM NaCl) and protease inhibitors (Roche, Penzberg, Germany). After 20 min of incubation on ice, lysates were centrifuged at 14,000 rpm at 4 °C for 20 min. The supernatants were transferred to new tubes, and the protein concentrations were measured with a Bradford protein assay (Bio-Rad Laboratories, Munich, Germany). A total of 20 µg of each cell lysate was resolved on 6–10% bis-tris/acrylamide gels and blotted onto nitrocellulose membrane (GE Healthcare, Düsseldorf, Germany), followed by blocking for 1 h in 5% nonfat milk and an incubation with primary antibodies at 4 °C under constant shaking overnight. The membrane was washed in TBS/0.05% Tween-20, incubated with HRP-coupled secondary antibodies at room temperature for 2 h, and, finally, developed with ECL. Signals were quantified on a Vilber Fusion FX6 (Marne-la-Vallée, France).

Immunoprecipitations were performed using activated CNBr beads (Sepharose 4 Fast Flow, GE Healthcare). Then, 0.5 g beads were solved in 5 mL 1 mM HCl and incubated at RT for 20 min. Beads were centrifuged at 3000× *g* for 1 min and washed 15 times. A subclass-specific mouse anti-rat IgG2b antibody (TIB 174/RG7/11.1, ATCC) was coupled to the beads (2 mg antibody in coupling buffer 0.3 M NaHCO_3_, 1.5 M NaCl, pH 8.3) at RT for 1 h. After that, beads were washed in coupling buffer and all unspecific binding sites were blocked with ethanolamine (1 M) at RT for 2 h. After washing in wash buffer (100 mM Tris/HCl, 0.5 M NaCl, pH 4.0) and in NaOAc buffer (0.1 M NaOAc, 0.5 M NaCl), beads were resuspended in PBS and used for coupling with EX-02 or isotype control antibodies. Therefore, 500 µL hybridoma supernatants were incubated with 60 µL anti-subclass specific beads at 4 °C overnight. Antibody-coupled beads were washed in PBS and incubated with 1 mg cell lysate at 4 °C overnight, then washed thrice in RIPA buffer with protease inhibitors. Finally, beads were pelleted by centrifugation, the supernatants were discarded, and beads were resuspended in 3x Laemmli buffer. After a final centrifugation at 1000× *g* for 5 min, the supernatant was used for PAGE and Western blot analysis.

Pull-down of truncated SRRM2 proteins and immunoblotting were carried out using whole-cell lysate prepared from respective HAP1 cell lines (as published in Ilik et al. 2020). Approximately 10 million cells were resuspended with 600 µL of 1 × NLB + 1 × Complete Protease Inhibitor Cocktail + 1 × PhosSTOP, and kept on ice for 15 min. The lysate was cleared by centrifugation at ~20,000 *rcf* for 10 min at 4 °C. Clarified lysate was split into two tubes; to one tube, 25 µL (slurry) of GFP-trap agarose beads were used (Chromotek, Munich, Germany), incubations were carried out overnight in the cold-room. Bound proteins were eluted with 50 µL of 1xLDS sample buffer (ThermoFisher Scientific, NP0007) + 100 mM beta-mercaptoethanol at 80 °C for 10 min and run on 3–8% Tris-acetate gels (Thermo Fisher Scientific, EA0375PK2). Gels were run at 80 V for 3 h and transferred onto a PVDF membrane in Tris-Glycine buffer for 90 min at 90 V. Antibody incubations were conducted at 4 °C overnight.

### 2.4. SRRM2 His Fusion Protein

HEK293 cells were transiently transfected with an expression plasmid encoding AA 1889-2150 of SRRM2 as a His fusion protein (herein referred to as tr04-His). Then, 48 h later, cells were lysed in urea buffer (8 M urea, 0.1 M NaH_2_PO_4_, 10 mM Tris-HCl, 0.05% Tween-20, 20 mM imidazole) and centrifuged at 4500 rpm for 10 min. Supernatants were incubated with Ni-NTA agarose beads (Qiagen, Hilden, Germany) overnight. The suspension was again centrifuged at 2000 rpm for 5 min, and the supernatant was discarded. Beads were washed twice in lysis buffer and proteins were finally eluted with lysis buffer containing 0.5 M imidazole and were stored at −20 °C.

### 2.5. Confocal Microscopy

Cells were plated on a cover slips in 12-well plates in normal media overnight to 70% confluency. The cells were washed once with PBS, fixed using 4% PFA for 10 min, followed by a permeabilization step with 0.3% Triton-X 100 for another 10 min. Antigen blocking was performed with 3% bovine serum albumin (BSA) in PBS for 40 min. All further antibody dilutions were prepared in PBS containing 1% BSA and 0.1% Triton-X 100. Subsequently, cells were incubated for 1 h with EX-02 (diluted to 1 µg/mL) and for surface staining either with EpCAM clone C215, diluted to 1 µg/mL, or integrin-α3 (Santa Cruz, Heidelberg; clone sc13545, diluted 1:250) antibodies. The sample was then washed three times followed by an incubation for 1 h with the following secondary antibodies: goat anti-rat Alexa Fluor 647 antibody (ThermoFisher Scientific, #A78947, 1 µg/mL), goat anti-mouse Alexa Flour 488 antibody (ThermoFisher Scientific, #A-11001, 1 µg/mL), and DAPI (diluted 1:10,000). After washing four times, cover slips were mounted using PoLong Gold antifade mountant (ThermoFisher Scientific, #P36930) Cells were imaged with a Leica TCS SP8 laser scanning microscope system. For surface staining, cells were first blocked, then fixed and permeabilized as described above. Afterwards, an extra staining step with DAPI (diluted 1:10,000) was performed for 10 min.

### 2.6. CAR Design and Viral Vector Production

Sequences of the rat-derived SRRM2-specific antibody EX-02 were synthesized in V_H_→V_L_ and V_L_→V_H_ orientations and cloned into a lentiviral expression vector carrying a second-generation CAR backbone with 4-1BB/CD137 and CD3-ζ transactivation domains. The integrity of the CAR was confirmed by sequencing. Lentiviral vectors were produced by cotransfecting HEK293 cells with VSV-G envelope; GAG-Pol packaging factors, and Rev plasmids. Supernatants were harvested 72 h later, filtered, concentrated by centrifugation, tested on HEK293 cells to calculate the titer of infectious virus, and, finally, cryopreserved.

### 2.7. Manufacturing and In Vitro Testing of CAR T-Cells

Fresh or thawed PBMCs were resuspend in PRIME-XV medium (Fujifilm, Ratingen, Germany). Cells were counted and adjusted to 2.0 × 10^6^/mL. Anti-CD3 (Miltenyi Biotec, Bergisch-Gladbach; Germany) and anti-CD28 (Miltenyi, Bergisch Gladbach, Germany) antibodies were added to a final concentration of 100 ng/mL, and cells were cultured with IL-7 and IL-15 (AcroBiosystems, Beijing, China) both at final concentrations of 10 ng/mL at 37 °C for 24 h. EX-02 CAR lentiviral vector (Genscript, Nanjing, China) was added at a calculated multiplicity of infection (MOI) of 5.0. Medium was replaced by PRIME-XV medium containing IL-7 and IL-15 24 h later, and the cell concentration was adjusted to 0.8 × 10^6^/mL. After four days of culture, samples were analyzed by flow cytometry using a BD Canto (BD Biosciences, Heidelberg, Germany) and analyzed using FlowJo (Tree Star, Ashland, OR). CD3, CD4, CD8, and CD45 antibodies (ThermoFisher Scientific) were used to characterize the human T-cell population; CAR expression on transduced cells was detected with an anti-rat IgG (H+L) F(ab′)_2_ fragment Alexa647 (#712-606-150; Jackson ImmunoResearch, Cambridge, UK). Cells were harvested on day six of culture and used for functional assays.

For this, 50,000 adherent cancer target cells per well were seeded into E-plate 16 chambers, and EX-02CAR-T or mock-T cells were added 24 h later. Target cell lysis was continuously measured with a xCELLigence live cell analysis system (OMNI Life Science, Bremen, Germany). Alternatively, target cells and T-cells were mixed in 96-well plates and incubated for 48 h. Afterwards, the supernatants were collected and IFN-ɣ and IL-2 concentrations were measured with ELISA Flex human (ALP) kits (Mabtech, Nacka Strand, Sweden), and dead target cells were identified with Zombie Green (Biolegend, Amsterdam, The Netherlands), according to the manufacturer’s protocol.

### 2.8. CRISPR/Cas9

CRISPR guide (g)RNA sequences were designed using the IDT CRISPR HDR design tool. Oligonucleotides, ordered from IDT (Dessau, Germany), were annealed to Cas9 protein (IDT) and the ribonucleoprotein complex was transfected into SKOV-3 cells using a 4D-Nucleofector (Lonza, Basel, Switzerland). Bulk transfected cells were checked by flow cytometry for successful knockdown, and clonal cell lines were obtained by single-cell cloning. Successful targeting of the genomic sequence was verified by PCR amplification and sequencing. The following gRNAs were used: Hs.Cas9.SRRM2.1.ACAGGATTAGGCCGCTTCACCA, Hs.Cas9.SRRM2.1.AQ: GTAAGAATCACTGATGCCAA.

### 2.9. Immunohistochemistry

Immunohistochemistry on 5 µm sections from frozen or paraffin-embedded tissues was performed using the biotin-peroxidase method (Vector laboratories, Burlingame, CA, USA). IHC staining and scoring were evaluated by a trained pathologist.

### 2.10. Mass Spectrometry

Eluted proteins from IPs were proteolyzed with LysC and trypsin with filter-aided sample preparation procedure as described in [21]. Acidified eluted peptides were analyzed on a Q Exactive HF mass spectrometer (ThermoFisher Scientific) in the data-dependent mode. Approximately 0.5 µg peptides per sample were automatically loaded to the online coupled ultra-high-performance liquid chromatography (UHPLC) system (Ultimate 3000, ThermoFisher Scientific). A nano trap column was used (300-µm ID X 5 mm, packed with Acclaim PepMap100 C18, 5 µm, 100 Å; LC Packings, Sunnyvale, CA) before separation by reversed phase chromatography (Acquity UHPLC M-Class HSS T3 Column 75 µm ID X 250 mm, 1.8 µm; Waters, Eschborn, Germany) at 40 °C. Peptides were eluted from the column at 250 nL/min using increasing ACN concentration (in 0.1% formic acid) from 3% to 41% over a linear 95 min gradient. The normalized collision energy was 27, and the spectra were recorded in profile mode.

The raw files were loaded to the Progenesis QI software (version 4.1, Waters) for label-free quantification and analyzed as described in [22,23]. MS/MS spectra were exported as Mascot generic file and used for peptide identification with Mascot (version 2.4, Matrix Science Inc., Boston, MA, USA) in the SwissProt Human protein database (release 2017_02, 20,237 sequences). Search parameters used were as follows: 10 ppm peptide mass tolerance and 0.6 Da fragment mass tolerance; one missed cleavage allowed, carbamidomethylation was set as fixed modification; methionine oxidation and asparagine or glutamine deamidation were allowed as variable modifications. A Mascot-integrated decoy database search was included, and peptide assignments were filtered for a Mascot percolator ion score cut-off of 13 and a significance threshold of *p* < 0.01. Peptide assignments were reimported into the Progenesis QI software, and the abundances of all unique peptides allocated to each protein were summed up. The resulting normalized abundances of the individual proteins were used for calculation of fold-changes of protein ratios between eluates from clone EX-02 and an isotype control. Proteins with at least 2 unique peptides and spectral counts in the EX-02 IP and no spectral counts in the isotype control were considered.

### 2.11. Xenograft Studies

Female NOG mice, SPF grade, were purchased from Beijing Charles River Laboratory Animal Technology Co., Ltd. (laboratory animal quality certificate number 110011231110590332). Mice were raised in the facility of Ruiye Model Animal (Guangzhou, China) Biotechnology Co., Ltd. (experimental animal use license number SYXK (Guangdong) 2020-0218). The design of the experimental study was approved by the Laboratory Animal Ethics Committee of Ruiye Model Animal (Guangzhou) Biotechnology Co., Ltd. (approval number: RYEth-20231026384).

BxPC-3-luc cells, purchased from Guangzhou Yuanjing Biotechnology Co., Ltd., were cultured in DMEM with 10% FBS. Cells were counted, resuspended in DMEM, and then 300,000 cells per mouse were injected intraperitoneally in a final volume of 100 µL. No toxicity related to the EX-02CAR-T cells was observed, i.e., body weight, habitus, and behavior were normal. Additionally, macroscopic postmortem inspections did not reveal any indication for tissue and organ damage. Tumor growth rates were monitored by longitudinal measurements of whole-body bioluminescence signals using the IVIS Lumina Series III imaging system equipped with a camera box and warming stage (PerkinElmer Inc., Waltham, MA, USA). Briefly, mice were anesthetized with isoflurane and fixed in the imaging chamber. Then, 3 mg/mouse of D-Luciferin potassium salt (#122799; PerkinElmer Inc.) were injected intraperitoneally 17 min before imaging. Total flux values were determined from the regions of interest (ROIs) covering the entire abdomen of each mouse and were presented in photons (p)/second (s), using Living Image software (PerkinElmer Inc., Boston, MA, USA).

### 2.12. Isolation of EVs

EVs from permanent cancer cell lines were isolated as described previously [24]. Briefly, cells were kept in EV-depleted medium for 3 days, and conditioned supernatants were centrifuged at increasing forces (10 min at 300× *g*, 4 °C, and 20 min at 5000× *g*, 4 °C) to remove cells and debris. After filtration of the supernatant (pore size 0.45 µm), EVs were then precipitated by ultracentrifugation (100,000× *g*, 2 h). Pellets were resuspended in PBS and further purified by flotation into a 4 mL iodixanol (Optiprep^®^, Sigma Aldrich, Taufkirchen, Germany) gradient. Ten fractions of 400 µL each were collected from the top, and their specific densities were measured with a refractometer (Reichert Technologies, Unterschleißheim, Germany). Particle concentration of EV samples was determined by nanoparticle tracking analysis (NTA) using the ZetaView PMX110 instrument (Particle Metrix, Inning, Germany) and the corresponding software (ZetaView 8.02.31).

### 2.13. EV Sandwich ELISA Assay

ELISA plates were coated overnight with antibodies diluted in PBS to a final concentration of 1 µg/mL. The next day, plates were washed 4 times with ELISA washing buffer (PBS/0.05% Tween-20) and incubated with blocking buffer (1% milk powder in PBS) at room temperature for 2 h. EV samples were diluted in blocking buffer to a concentration of 4 * 10^9^ EVs/mL, and 50 µL/well were added for 2 h in triplicates. Wells were then incubated with an HRP-coupled CD63-specific antibody (1 µg/mL) and developed with OptEIA TMB substrate solution (BD Biosciences, Heidelberg, Germany), stopped with 1 M H_2_SO_4_, and read immediately on a plate reader at 450 nm.

### 2.14. Statistical Methods

All statistical analyses were performed using R-4.3.2. To capture the individual-specific mouse effects and to account for the repeated measurements, a linear mixed model was performed using the lmer function from the lme4 package in R. We compared the measurements for three groups: EX-02CAR-T, Mock-T, and PBS. Mock-CAR-T was used as the reference group. Each group consisted of five mice, which were studied across six different time points. In total, we had 30 observations per group.

## 3. Results

### 3.1. An Antibody That Is Specific for SRRM2 Binds to the Surface of Living Cancer Cells

From the immunization of a rat with a mixture of EVs isolated from conditioned supernatants of SKOV-3, Capan-1, T-47D, and HeLa cells, representing four major types of cancer, we obtained a series of hybridomas that we first tested by flow cytometry for binding to the cell lines of origin. We identified hybridomas that bound to one, two, or even all four cell lines, and these latter hybridomas were selected for more detailed investigations. First, their specificities were identified by immunoprecipitations using cell lysates as a source for the target proteins, followed by mass spectrometric analysis. Among other specificities, we identified antibody 23A7 (hereafter referred to as EX-02), which specifically precipitated the nuclear protein SRRM2 (Figure 1A). This result was unexpected because SRRM2 has so far only been described in the literature as a nuclear speckles protein [5] that can accumulate in the cytoplasm of diseased neurons in certain neurodegenerative diseases [10,11,12]. Yet, our flow cytometry data clearly evidenced that the protein is highly expressed on the surface of various human cancer cell lines (Figure 1B), while primary normal hepatocytes and PBMCs from healthy volunteers stained completely negative (Appendix A). Subsequent immunoblots with lysates from different cancer cell lines substantiated the assumed specificity of EX-02 as it gave a specific signal at approximately 300 kDa, corresponding to the expected size of SRRM2 (Figure 1C). Interestingly, the antibody also detected a minor band of approximately 200 kDa in size, pointing to the existence of different SRRM2 isoforms present in cancer cell lines at various ratios. As an additional proof of specificity of EX-02, we generated a CRISPR/Cas9-driven knockout of SRRM2 in SKOV-3 cells that resulted in complete loss of EX-02 surface binding (Figure 1D). Collectively, these data prove that EX-02 is an SRRM2-specific antibody.

### 3.2. EX-02 Binds within a Fragment Comprising Amino Acids 1889–2150 of SRRM2

To identify the binding region of EX-02 within SRRM2, we made use of a series of HAP1 cell lines carrying endogenously GFP-tagged SRRM2 with C-terminal truncations at 11 different positions, which we described previously [5]. In brief, we generated cell lysates from these cell lines, immunoprecipitated endogenously expressed SRRM2 with various truncations using GFP-trap beads and analyzed the eluates by immunoblotting. Compared to blots using a polyclonal antibody against GFP serving as a loading control, EX-02 recognized the SRRM2 protein as far as truncation 3 (tr03), but no signal was detected from truncation 4 (tr04) onwards (Figure 2A). We, therefore, concluded that EX-02 binds to a region comprising amino acids (AA) 1889-2150 of SRRM2. To verify this result, we cloned this fragment (hereafter referred to as tr04) and expressed it as His fusion protein in HEK293 cells. Immunoprecipitations of tr04-His with EX-02 from lysates of transfected HEK293 cells, followed by SDS-PAGE and an immunoblot with an His-specific antibody, unambiguously showed that EX-02 specifically precipitated tr04 (Figure 2B). EX-02 also gave a specific signal on immunoblots at the size of tr04 in transfected, but not in nontransfected, cells (Appendix A).

Additionally, we tested EX-02 in sandwich ELISA assays. For this, we coated 96-well polystyrene plates with an anti-His antibody and then incubated them with purified tr04-His protein or a control protein (MISP-His). Plates were then incubated with EX-02 followed by incubation with an HRP-coupled anti-rat antibody. Finally, plates were developed with TMB substrate and analyzed photometrically (Figure 2C). A clear signal was obtained for tr04-His but not for MISP-His, indicating that EX-02 binds to the fragment tr04 of SRRM2, thus confirming the results described above. Altogether, these experiments corroborated the specificity of EX-02 and, thus, the surface localization of SRRM2 on cancer cells.

### 3.3. Confocal Microscopy Implies the Existence of Diverse Isoforms of SRRM2

The identification of SRRM2 on the surface of cancer cells was unexpected because the protein has thus far been identified as a key component of nuclear speckles. However, the protein has also been found to accumulate in the cytoplasm of neurons in neurodegenerative diseases [10,11,12]. To verify the results described above, we performed confocal microscopy and compared EX-02 with commercial SRRM2 antibodies. In the first series of experiments, we stained living Capan-1 and HeLa cells with different SRRM2 antibodies, followed by fixation and incubation with suitable fluorochrome-labeled secondary antibodies. Whereas EX-02 gave a clear signal located at the plasma membrane, as evidenced by colocalization with an antibody targeting the surface protein EpCAM, the commercial SRRM2-specific antibodies used did not yield a detectable signal under these experimental conditions (Figure 3A). Flow cytometric analysis of HeLa cells with the SRRM2 antibodies mentioned above and two additional commercial antibodies corroborated that only EX-02 binds surface SRRM2 (Appendix A).

Because the commercial SRRM2 antibodies are, in principle, suitable for immunofluorescence microscopy, we next performed confocal microscopy as described above, but we applied the first antibody after PFA fixation and permeabilization to also allow for intracellular staining. This time, all three antibodies used stained SKOV-3 cells intracellularly, interestingly with diverse staining patterns (Figure 3B). We observed a strict nuclear staining with the Abcam antibody #122719 which, according to information from Abcam’s technical support, had been generated by immunizations with “a proprietary peptide located within the region ranging from AA #1000-1270 of SRRM2”. In sharp contrast, the polyclonal Biozol antibody, derived from immunization with a fragment comprising AA #2037-2087 of SRRM2, and, thus, in the same region as EX-02, almost exclusively stained the cytoplasm of SKOV-3 cells faintly. As on living cells, the EX-02 antibody preferentially stained the plasma membrane with some weak staining of the cytoplasm. Similar results were obtained with A549 lung cancer and UWB1.289 ovarian cancer cells (Appendix A). Due to these diverse staining patterns, it is tempting to speculate that different post-translationally modified isoforms of SRRM2 exist in cancer cells and that these modifications account for the divergent intracellular localizations of the protein. We also concluded that EX-02 binds an epitope of SRRM2 that is exposed at the cell surface, whereas the commercial antibodies tested bind to epitopes that are not accessible on living cells or that are modified in surface-tethered SRRM2 in a way that impedes antibody binding.

### 3.4. SRRM2 CAR-T Cells Are Reactive against Cancer Cells In Vitro

Chimeric antigen receptor (CAR)-T cells have emerged as promising immunotherapeutic tools, particularly against hematologic malignancies. To explore the reactivity and specificity of SRRM2 CAR-T cells, we designed lentiviral vectors encoding single-chain variable fragments (scFV; in V_H_→V_L_ and V_L_→V_H_ orientations) based on the variable heavy and light chain sequences of the EX-02 antibody fused to a second-generation CAR backbone construct, carrying the CD3-ζ and 4-1BB activation domains. We produced viral vectors in HEK293 cells and used them to transduce primary T-cells that were preactivated for two days with plate-bound CD3 and CD28 antibodies. During the development of this transduction protocol, we repeatedly observed a higher transduction and CAR expression rate with the EX-02 V_H_→V_L_ construct, which we therefore used for the subsequent experiments. T-cells transduced with this construct are hereafter referred to as EX-02CAR-T cells.

As a functional test, we incubated EX-02CAR-T effector cells with SKOV-3 and Capan-1 target cells at E:T ratios of 10:1, 5:1, and 2:1 and measured T-cell killing over time in a xCELLigence system and quantified IFN-ɣ secretion in ELISA assays. We observed efficient target cell killing (Figure 4A) and concomitant high IFN-ɣ levels specifically in cocultures with EX-02CAR-T cells, but not with mock-T cells (Figure 4B and Appendix A).

### 3.5. Development of a CAR-T Xenograft Model

To gain insight into the activity of EX-02CAR-T cells in vivo, we set up a human xenograft model in immunocompromised NOG mice to mimic peritoneal metastasis. To this end, a total of 300,000 BxPC-3-luc pancreatic cancer cells per mouse were injected intraperitoneally, and successful engraftment was confirmed by bioluminescence imaging one day later. At that timepoint (=day 0), mice (n = 5 per group) were injected with either 4 × 10^6^ EX-02CAR-T cells (transduction rate 73%) or mock-T cells in a final volume of 100 µL PBS intraperitoneally. Control animals received the same volume PBS as vehicle control only. The injection of the human T-cells was well tolerated, and no adverse events were observed. To follow the growth of the xenografted BxPC-3-luc cells over time, bioluminescence imaging was performed at days 5, 8, 12, 19, and 26 after T-cell injection (Figure 5A,B). These measurements revealed that EX-02CAR-T cells had a statistically significant inhibitory activity as compared with mock-T cells, and both groups differed significantly from the PBS control group. At day 26, when the experiment had to be terminated for animal welfare reasons, animals treated with EX-02CAR-T cells reached an average tumor burden of 5.3 × 10^8^ light units (p/s), as compared to 2.6 × 10^9^ p/s for the mock-T cell treated mice (*p* = 0.014) and 3.6 × 10^9^ p/s for mice of the PBS group (*p* < 0.01) (Figure 5C). Mice treated with EX-02CAR-T cells also revealed a significantly higher CD8^+^/CD4^+^ ratio in the blood and the spleen as compared to mock-T treated animals, which has been shown to be predictive for CAR-T efficacy and a low risk of treatment failure [25,26] (Appendix A).

### 3.6. Patient-Derived Carcinomas Show Strong Membrane SRRM2 Staining

To obtain a first insight into the expression profile of surface-SRRM2 in vivo, we performed immunohistochemistry (IHC) on various types of primary cancer and corresponding adjacent tissues. Overall, we tested 157 tumor samples derived from bile duct, ovary, pancreas, and stomach. Figure 6A shows the typical staining of a stage 3c ovarian serous carcinoma and the corresponding adjacent tissue, while IHC staining of other types of carcinomas with EX-02 are shown in Appendix A. The table in Figure 6B provides an overview of the number of cases stained alongside IHC scores as they are used for the definition of Her2/neu staining levels [27]^.^ Overall, we observed moderate to strong membrane staining in most cancers, whereas almost all adjacent tissues investigated stained negative. Only a few (4/37 tissues adjacent to bile duct carcinoma and 1/48 adjacent to a pancreatic carcinoma) revealed detectable membrane staining. In parallel, we also stained a small number of squamous cell carcinomas of the head and neck region (HNC) because some of the HNC-derived cell lines revealed surface SRRM2 expression (Appendix A). Interestingly, and in contrast to the data described above, primary HNC mostly revealed a nuclear localization of SRRM2 (Appendix A). The reason for this discrepancy in SRRM2 location between gastrointestinal carcinoma described above and HNC is unknown. In essence, even though classical IHC cannot define the precise subcellular localization of membrane-associated SRRM2, it is tempting to speculate that SRRM2 is exposed on the surface of primary cancer cells like it is on most cancer cell lines.

### 3.7. SRRM2 Is Present on the Surface of Extracellular Vesicles

SRRM2 has previously been detected in cancer-derived EVs by proteomic analysis [28,29]. To confirm this result, we performed an immunoblot with EVs derived from different cell lines. As shown in Figure 7A, SRRM2 was present in EVs at detectable levels. Because this experiment did not allow us to define the exact localization of SRRM2 within EVs, we precipitated EVs from SKOV-3 cells by ultracentrifugation and further purified them via floatation into a density gradient. From this gradient, we obtained ten fractions, which we first tested in immunoblots for the presence of SRRM2 and the pan-EV markers CD63 and CD81. As shown in Figure 7B, all three proteins co-sedimented and were enriched in fractions 2 and 3 of the gradient, corresponding to a specific density of approximately 1.07–1.09 and, thus, within the buoyant density described for EVs. Calnexin, an integral protein of the endoplasmic reticulum membrane that is often used as an “exclusion marker” to demonstrate the purity of EV preparations, was only detected in the lysate of SKOV-3 cells, and not in EVs. A subsequent NTA analysis further substantiated the enrichment of vesicles in fractions 2 and 3 (black columns in Figure 7B). To demonstrate the presence of SRRM2 on the surface of EVs, we performed a sandwich ELISA with the ten gradient fractions using EX-02 or an isotype control antibody as capture antibodies and an HRP-coupled CD63 antibody for detection. As depicted by the red line in Figure 7B, we obtained the highest signals with fractions 2 and 3 of the gradient that also contain the highest number of EVs. Similar ELISA assays with EVs derived from other cancer cell lines gave comparable results (Figure 7C). We concluded that SRRM2 is present on the surface of cancer-derived EVs and, hence, constitutes a marker for detection of cancer-derived EVs and is, thus, a potential diagnostic and prognostic circulating biomarker. Because EVs comprise a mixture of vesicles of different origin, and the number of EVs as measured by NTA did not exactly correlate with the ELISA signals, we propose that SRRM2 is only present on a subfraction of EVs enriched in fraction 3 of the gradient.

## 4. Discussion

In an attempt to identify new tumor-associated tumor antigens and to simultaneously generate the corresponding antibodies, we immunized a rat with a mixture of EVs derived from permanent human cancer cell lines. We obtained, among other antibodies, EX-02, a monoclonal antibody that turned out to be specific for SRRM2, known as an RNA-binding spliceosomal protein and a major component of nuclear speckles. Yet, we unexpectedly observed that EX-02 also binds to living tumor cells, and additional investigations proved that SRRM2 is, indeed, exposed on the surface of cancer cells of different origin. Subcellular mislocalization of nuclear proteins, dysfunctional nuclear speckles, and irregular splicing events are hallmarks of various neurodegenerative diseases including Alzheimer’s disease, amyotrophic lateral sclerosis, frontotemporal dementia, and Parkinson’s disease. Cytoplasmic accumulation of nuclear proteins and their sequestration from the nucleus in pathological conditions have, for example, been described for RNA-binding proteins involved in RNA splicing like FUS and TDP-43 (see [30] for review). Also, SRRM2 has been found in cytoplasmic amyloid-like aggregates containing the tau protein [10], and it has been proposed that ß-amyloids or inflammatory events account for its mislocalization and cytoplasmic accumulation [31]. ERK1/2 has been identified as one kinase that phosphorylates SRRM2, and cytoplasmic aggregations of phosphorylated SRRM2 can even precede that of ß-amyloid and pathogenic tau [12]. However, the potential role of mislocalized SRRM2 (and other RNA-binding proteins) in the etiology and contribution to the pathology of neurodegenerative diseases remains to be elucidated. We are the first to identify SRRM2 on the surface of cancer cells, while others have described other RNA-binding proteins exposed on the cell surface previously. Tonapi et al. described the translocation of a spliceosomal complex consisting of at least 13 core components, including various member of the hnRNP family and paraspeckle-resident SFPQ (but not SRRM2), to the surface of non-Hodgkin’s lymphoma cells [32]. Borer et al. reported that nucleolin, which binds preribosomal RNA, constantly shuttles between the nucleus and the cytoplasm [33]^,^ and Joo et al. observed that phosphorylated nucleolin appears on the surface of various types of cancer cells [18]. Thus, the intracellular shuttling and translocation of nuclear RNA-binding proteins to the cytoplasm and plasma membrane seems to be a relatively frequent phenomenon in cancer cells, rendering these proteins attractive target molecules. In this context, it is important to mention that SRRM2 is exposed on many permanent cancer cell lines and is absent from normal cells, at least those we investigated. Even more importantly, various primary cancer cells also reveal SRRM2 membrane staining, while in adjacent noncancer cells the protein mainly shows the expected nuclear staining. Hence, surface SRRM2 is a new druggable (as demonstrated with CAR-T cells) target molecule on cancer cells. Additionally, SRRM2 is also present on the surface of cancer-derived EVs, implicating the use of the protein as an attractive circulating biomarker for EV-based diagnostic approaches.

Results from a first human xenograft model with pancreatic cancer cells and SRRM2-specific CAR-T cells corroborate the idea of SRRM2 as an attractive target molecule. Even though SRRM2 CAR-T cells were unable to eliminate the xenografted tumor cells completely, we observed that even a single injection of CAR-T cells showed a clear antitumor activity resulting in significant tumor growth delay. However, it should be mentioned that this is the first animal model so far, and that the antitumor activity and specificity of SRRM2 CAR-T cells should be further investigated.

Transport of intracellular proteins to the surface of cancer cells is an exciting phenomenon that has already been described for diverse proteins. Many of these proteins primarily function as chaperones within the ER lumen and membrane, but their role on the cell surface is less understood. Stress due to a lack of nutrients and oxygen and the unfolded protein response has been discussed to possibly trigger this intracellular translocation, but exactly why and how SRRM2 and other RNA-binding proteins are transported to and associate with the plasma membrane, their possible function, and their binding partners are currently largely unknown. Such investigations are currently ongoing in our group.

In conclusion, here, we described our approach for the identification of new target molecules on the surface of cancer cells and antibody generation using cancer-cell-derived EVs for immunizations. From such an immunization, we unexpectedly obtained an antibody targeting the nuclear speckles protein SRRM2 and demonstrated that SRRM2 is exposed on the surface of most cancer cell lines investigated. Even more relevant, primary tumor cells also clearly reveal a membrane localization of SRRM2. We therefore consider surface SRRM2 a new druggable tumor-associated target molecule and EX-02 an attractive first-in-class candidate for the development of targeted therapies. As such, our data warrant future investigations of the potential role of mislocalized intracellular proteins in general, and SRRM2 in particular, in the etiology and progression of cancer.

## Figures and Tables

**Figure 1 cells-13-01563-f001:**
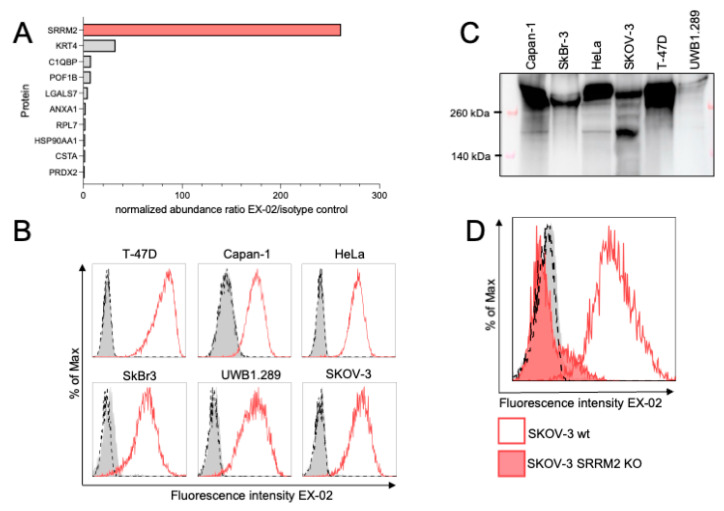
EX-02/23A7 is an SRRM2-specific antibody. (**A**) EX-02 specifically precipitates SRRM2 from SKOV-3 lysates. EX-02 coupled to CNBr beads was incubated with cell lysates from SKOV-3 cells overnight, pelleted by centrifugation, and precipitated, eluted with Laemmli buffer. Eluted proteins were identified by mass spectrometry as described. An isotype antibody specific for maltose binding protein was used as a control. (**B**) EX-02 binds to the surface of cancer cell lines as revealed by flow cytometry. Cells were incubated with EX-02, followed by incubation with an Alexa647-labeled anti-rat IgG secondary antibody. (**C**) On immunoblots, EX-02 detects a protein of approximately 300 kDa, corresponding to the calculated size of SRRM2. In some cell lines, it also detects a protein at approximately 200 kDa, possibly an SRRM2 isoform. (**D**) EX-02 binds to the surface of wildtype SKOV-3 cells, but not an SRRM2 CRISPR/Cas9 knockout clone.

**Figure 2 cells-13-01563-f002:**
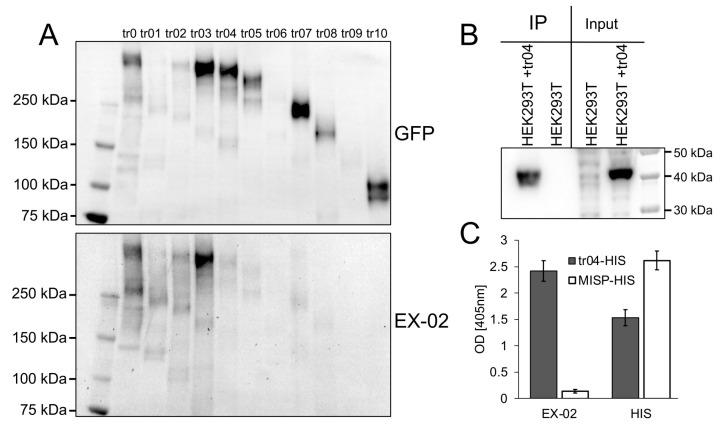
EX-02 binds to a fragment comprising amino acids 1950–2100 or SRRM2. (**A**) Lysates from HAP1 cell lines expressing truncated SRRM2-GFP fusion proteins [5] were incubated with anti-GFP trap beads, and the eluates were analyzed by immunoblotting with EX-02 or an anti-GFP antibody. (**B**) Lysates from parental HEK293 cells (HEK) and HEK cells transfected with an expression plasmid for tr04 were incubated with EX-02 coupled to CNBr beads. Precipitated proteins were eluted and separated by PAGE and blotted onto a PVDF membrane. The membrane was incubated with an HRP-coupled anti-His antibody and developed with ECL. Cell lysates were included as input control. (**C**) ELISA assay with EX-02 and an anti-His antibody. A 96-well polystyrene plate was coated with tr04-His or MISP-His as a control, blocked, and then incubated with EX-02 or an anti-His antibody to show comparable coating with tr04-His and MISP-His. Finally, plates were incubated with an HRP-coupled anti-rat antibody and developed with TMB.

**Figure 3 cells-13-01563-f003:**
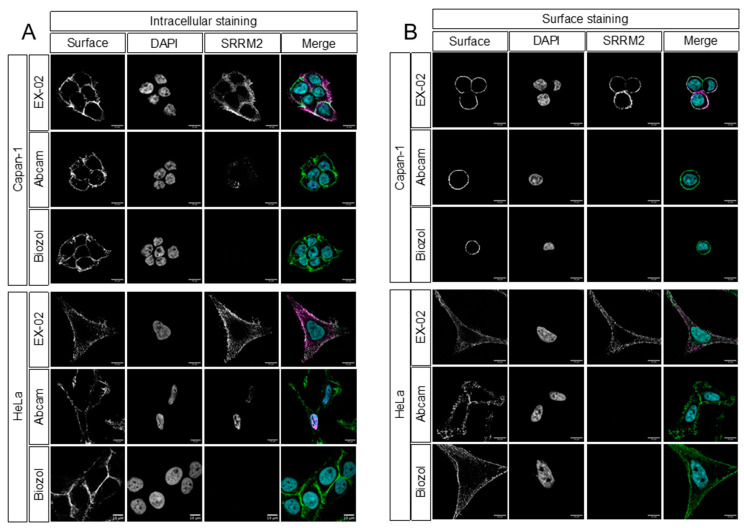
EX-02 binds to surface SRRM2. Confocal microscopy with (**A**) vital and (**B**) fixed and permeabilized Capan-1 and HeLa cells. EX-02, Abcam, clone #122719, and Biozol (MBS9609206) were used as SRRM2 antibodies (magenta); an EpCAM or IGF-α3 antibody were used to stain an established surface protein (green). Nuclei were counterstained with DAPI (cyan).

**Figure 4 cells-13-01563-f004:**
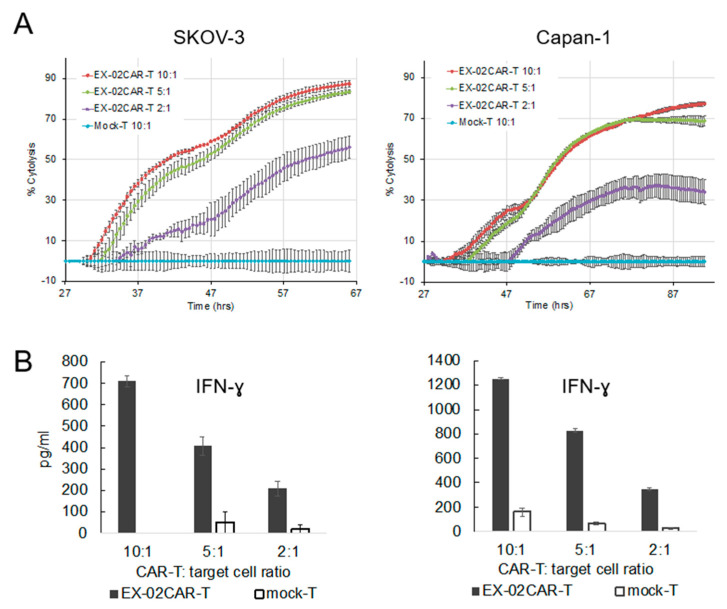
EX02CAR-T cells kill target cells in vitro. (**A**) SKOV–3 or Capan–1 cells were incubated with EX-02CAR-T cells at different E:T ratios. Killing over time was analyzed with an xCELLigence system. Mock-T cells were used as a negative control. (**B**) Supernatants from SKOV–3 or Capan–1 cells incubated with EX-02CAR-T cells were analyzed for IFN-ɣ and IL-2 with commercial ELISA assays.

**Figure 5 cells-13-01563-f005:**
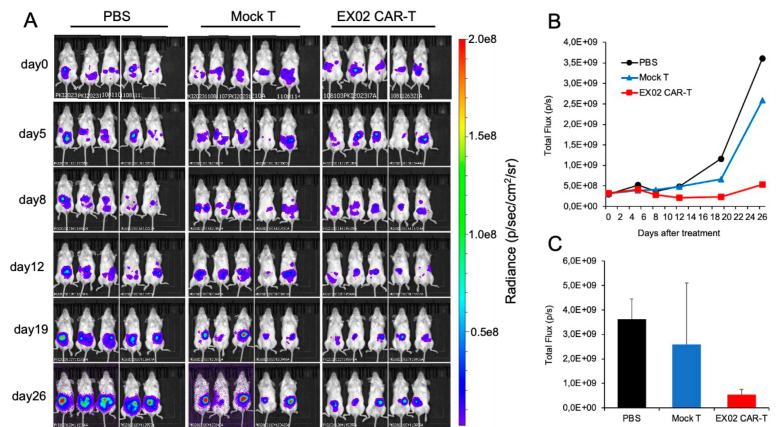
EX-02CAR-T cells inhibit tumor growth in a human xenograft model. (**A**,**B**) Bioluminescence imaging of xenografted BxPC-3-luc pancreatic cancer cells in mice treated with EX-02CAR-T cells, mock-T cells, or with PBS, performed at different time points after T-cell injection. (**C**) Total flux data at day 26 after treatment.

**Figure 6 cells-13-01563-f006:**
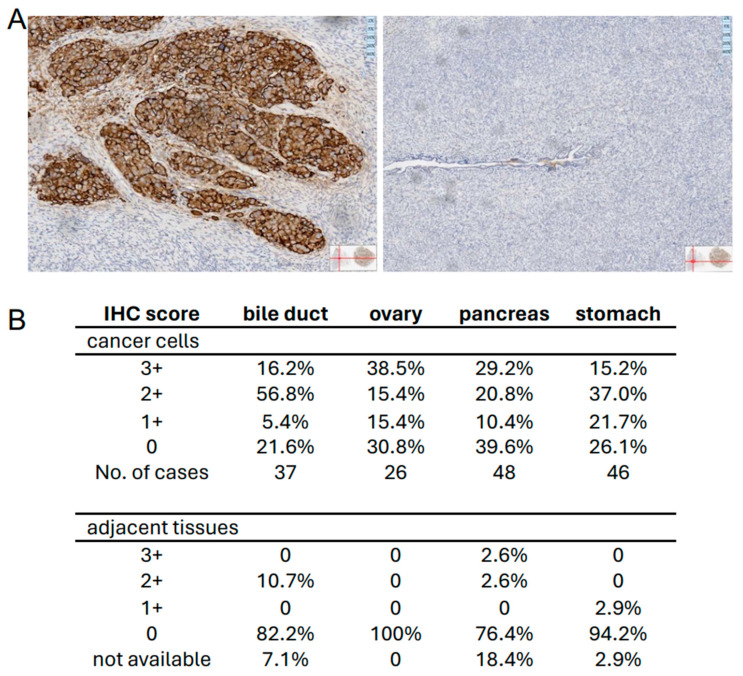
The surface localization of SRRM2 is cancer-associated. (**A**) IHC with EX-02 of an ovarian serous carcinoma, stage 3c (**left**), and adjacent normal ovarian tissue. (**B**) Overview of EX-02 membrane staining of different types of carcinomas and adjacent tissues. The IHC score used was the same as that used routinely for Her2/neu [27].

**Figure 7 cells-13-01563-f007:**
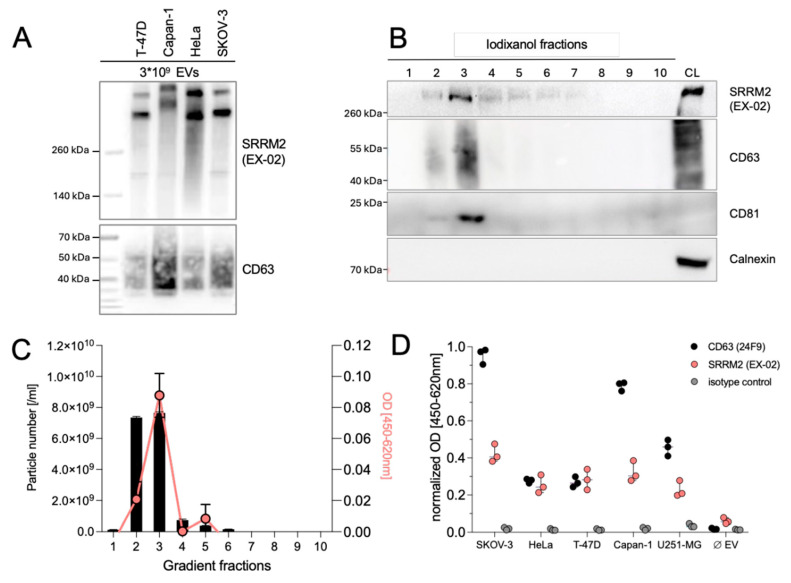
SRRM2 is present on the surface of cancer-derived EVs. (**A**) Conditioned supernatants of SKOV-3 cells were pelleted by ultracentrifugation and fractionated via an iodixanol density gradient. (**B**) Western blots revealed co-sedimentation and enrichment of SRRM2 with the pan-EV markers CD63 and CD81 in fractions 2 and 3 of the gradient, corresponding to specific densities of 1.07 and 1.09, respectively, which are typical for EVs. A lysate from SKOV-3 cells (CL) was included as a positive control. An antibody against the ER protein calnexin, that is excluded from EVs, was used to demonstrate the purity of the EV preparation. (**C**) NTA analysis (black columns) and sandwich ELISA of the ten fractions of the gradient using EX-02 and CD63-HRP as capture detection antibodies, respectively. (**D**) SRRM2 ELISA data of EVs derived from different cell lines.

## Data Availability

The data presented in this study are available on request from the corresponding author due to commercial interests.

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
