# Peer review of "The Nuclear Speckles Protein SRRM2 Is Exposed on the Surface of Cancer Cells"

_cells, 2024, doi:10.3390/cells13181563_

Round 1
Reviewer 1 Report
Comments and Suggestions for Authors
1. The manuscript contains a significant amount of duplication and requires extensive revision.
2. The title should be revised to more accurately reflect the specific findings of this scientific study.
3. The authors should provide the raw images of all the western blots included in the article.
Comments on the Quality of English LanguageAuthors need to re-write the manuscript to avoid the duplication and avoid grammatical errors.
Author Response
Comment 1: The manuscript contains a significant amount of duplication and requires extensive revision.
Response 1: We would like to thank the reviewer for this information. We have carefully re-read the manuscript and eliminated all repetitions that we noticed.
Comment 2: The title should be revised to more accurately reflect the specific findings of this scientific study.
Response 2: Although we feel that the current title is not 'wrong', we've changed it in order to more reflect the key findings of the manuscript, namely the surface localization of SRRM2 on cancer cells.
Comment 3: The authors should provide the raw images of all the western blots included in the article.
Response 3: These images have already been submitted to the journal and they are re-attached here.
Reviewer 2 Report
Comments and Suggestions for Authors
I am pleased to review the manuscript tilted as “The nuclear speckles protein SRRM2 is a new therapeutic tar-2 get molecule on the surface of cancer cells”. The research article summarizes the findings of experiments conducted using EVs for rat immunization and identified SRRM2 as a promising new target molecule exposed on the 28-cancer cell surface and showed that SRRM2-specific antibody can be used as a basis for the development of new targeted cancer therapies.
I do believe it’s a well written and well-constructed manuscript, showing the authors hard work and dedication. However, based on my review of the manuscript I would like to suggest some of the points for the overall improvement of the manuscript.
Following are my comments that I would like to be taken into consideration by the authors.
1. In the material and methods section, under the subheading generation of the antibodies the font style is changed and please make it consistent with the rest of the text.
2. Line 109 what was the time of the last centrifugation at 100000x g for EVs purification? Please mention in the text.
3. In results section Figure 1 please remove the red line under the different text in the figure such as red line under Hela.
4. Figure 7 also have the same red lines under the text please remove those.
5. The images quality if improved the paper would make so good initial impact.
6. Overall, the study is conducted in very detail. The conclusion is clearly supported by the experimental results and I feel very satisfied and agree to the publication of this article.
Best wishes….
Author Response
Comments 1: In the material and methods section, under the subheading generation of the antibodies the font style is changed and please make it consistent with the rest of the text.
Response 1: fixed
Comments 2: Line 109 what was the time of the last centrifugation at 100000x g for EVs purification? Please mention in the text.
Response 2: fixed
Comments 3: In results section Figure 1 please remove the red line under the different text in the figure such as red line under Hela.
Response 3: fixed
Comments 4: Figure 7 also have the same red lines under the text please remove those.
Response 4: fixed
Comments 5: The images quality if improved the paper would make so good initial impact.
Response 5: We've tried to improve the quality of the figures as good as possible.
Comments 6: Overall, the study is conducted in very detail. The conclusion is clearly supported by the experimental results and I feel very satisfied and agree to the publication of this article.
Reply 6: Thank you very much
Reviewer 3 Report
Comments and Suggestions for Authors
Kellner et al studied the potential of cancer cell surface SRRM2 as potential anticancer therapeutic target. The findings are interesting but the below requires further to be addressed.
Major comments:
- Please provide the rationale of using extracellular vesicles for immunization;
Line 317: the comparison was conducted between human cancer cell lines and primary hepatocytes/PBMCs. They are of different origins - please explain the reason for not testing in the cells of the same origin/organ.
Minor comments:
Line 308: please include the flow cytometry analysis as supplementary files.
Figure 1D: caption missing.
Figure 6B: It would be useful to include the N numbers and the p values here.
Comments on the Quality of English Language
readable; please have it proofread to minimize grammatical and spelling errors.
Author Response
Comments 1: Please provide the rationale of using extracellular vesicles for immunization
Reply 1: Our rational for using extracellular vesicles for immunization is already mentioned in the introduction (lines 47-50): Since in term of their composition EVs largely represent the cells of origin, we speculated that cancer derived EVs also carry proteins not yet recognized as TAAs, thus constituting a valuable tool for the identification of new druggable target molecules on the surface of cancer cells.
Comments 2: Line 317: the comparison was conducted between human cancer cell lines and primary hepatocytes/PBMCs. They are of different origins - please explain the reason for not testing in the cells of the same origin/organ.
Reply 2: single cell suspensions suitable for flow cytometry and derived from normal organs and tissues are very hard to get. We were lucky to have the chance to have access to normal hepatocytes. In order to compare tumor and corresponding normal tissues, we instead did a lot of immunohistochemistry (e.g. Figure 6).
Comments 3: Line 308: please include the flow cytometry analysis as supplementary files.
Reply 3: From this immunization, we obtained about 50 hybridoma which we screened by flow cytometry for their binding to 4 different cell lines. Some of them were binding to all 4 cell lines, others to only 1 or 2, and even others to none of them. Besides antibody 23A7, no other hybridoma is mentioned in this manuscript and thus considered not relevant. In addition. the FACS histograms are just 'standard' showing the binding of an antibody to a particular cell line, only. We thus feel that just a figure is neither informative nor relevant.
Comments 4: Figure 1D: caption missing
Reply 4: fixed
Comments 5: Figure 6B: It would be useful to include the N numbers and the p values here
Reply 5: The N numbers are mentioned (No. of cases). We quite don't understand which 'p values' the reviewer has in mind here. This figure lists the fraction of surface-SRRM2 positive primary tumors of different origine. We don't know what we should compare here to get a 'p value'.
Round 2
Reviewer 3 Report
Comments and Suggestions for Authors
comments are addressed
Comments on the Quality of English Languageokay